# Use of Personal Listening Devices and Knowledge/Attitude for Greater Hearing Conservation in College Students: Data Analysis and Regression Model Based on 1009 Respondents

**DOI:** 10.3390/ijerph17082934

**Published:** 2020-04-23

**Authors:** Sunghwa You, Chanbeom Kwak, Woojae Han

**Affiliations:** 1Division of Speech Pathology and Audiology, College of Natural Sciences, Hallym University, Chuncheon 24252, Korea; shyouuu@gmail.com (S.Y.); cksqja654@gmail.com (C.K.); 2Laboratory of Hearing and Technology, Research Institute of Audiology and Speech Pathology, College of Natural Sciences, Hallym University, Chuncheon 24252, Korea

**Keywords:** personal listening device, PLDs, college students’ recreational noise exposure, hearing conservations

## Abstract

Given the concern regarding increased hearing loss in young people who use personal listening devices (PLDs), the present study analyzes the experience of PLDs among college students to identify their knowledge of and attitude toward hearing conservation. It also explains their relationship between knowledge of hearing loss and attitude-related hearing conservation as a questionnaire response using a regression model. A total of 1009 Korean college students responded to an online questionnaire. As a survey tool, the Personal Listening Device and Hearing Questionnaire was adapted as a Korean version with 78 modified items under 9 categories. Using principal component analysis, specific factors were extracted, and their relationships and paths were confirmed using multiple regression analysis. The results of the knowledge category of the questionnaire indicate that most respondents knew how to maintain healthy hearing and understood the signs of hearing loss. Regardless, many college students habitually use PLDs at high levels in noisy environments; they do not recognize how to prevent hearing loss. Even though they continue their current use pattern for PLDs, they also had a positive attitude toward receiving more information about hearing conservation. According to the regression model, the students’ self-reported hearing deficits were due to the volume rather than the frequent use. Interestingly, knowledge about hearing loss may encourage students to develop a positive attitude toward reasonable restriction of PLD use. When PLD users have detailed knowledge about the hearing loss provided by professionals, we believe that most will avoid serious hearing problems and its risks and maintain a judicious attitude toward their own conservation.

## 1. Introduction

Contemporary researchers have offered evidence of hearing loss from exposure to loud recreational noise by focusing on groups of choristers [1], symphony orchestra musicians [2], attendees at rock concerts and discotheques [3], and users of personal listening devices (PLDs) [4]. The relationship between the use of PLDs and hearing damage has been a serious concern, especially for younger people [5]. For example, Ahmed et al. (2007) reported that as many as 82% of students currently have PLDs [6] and Portnuff et al. (2009) showed that young people, compared to older people, more frequently listen to music at a higher volume while not realizing that this intensity could potentially be hazardous [7]. Why are young people addicted to PLDs? These devices allow users to enjoy loud music without interfering with others [8], and they can serve as a companion during leisure hours and/or study time [9]. However, unlimited preferred listening levels (PLLs) increased significantly when using PLDs in noisy environments [10] and coupling with an earphone of the earbud type did not effectively attenuate background noise levels, thus greatly increasing the volume [11]. Furthermore, given the rapid development of electronic technology, access to PLDs has increased across all age groups, in addition, PLD users have changed the device they use from a portable CD player, MP3 player, or iPod [9] to a mobile phone (or Smartphone) with a free MP3 player function [12]. This new direction suggests the need to re-examine the use of PLDs among those young people who prefer to use the new technology and identify their knowledge of and attitudes toward hearing conservation.

Although a typical method used in previous PLD studies was to measure PLL and/or maximum output levels of commercially available PLDs, it is risky to infer hearing loss only induced from their use [13]. When we consider the output results of the PLDs from many studies, users’ listening levels may cause a certain amount of hearing insult because the levels exceed the criteria suggested by institutes like the National Institute for Occupational Safety and Health (NIOSH) and the Occupational Safety and Health Administration (OSHA); the values are 68.3~84.3 dBA for average PLL and 96.7~107 dBA for the maximum level [11,14]. However, Rice et al. (1987) disagreed in that only 5% of their participants were potentially exposed to dangerous levels of noise [15]. Turunen-Rise et al. (1991) also argued that most listeners used PLDs below the dangerous levels for a limited time, a behavior that might minimize hearing loss [16]. This discrepancy between experimental results seems to have stemmed from a small segment of the target population and different settings, times, or cultures by which to determine the users’ potential risk, and that issue warrants further study using a large data set and regional characteristics.

Based on World Health Organization (WHO) guidelines [17], the overuse of personal listening devices among college students should be restricted by reducing either the user’s time or the volume of the PLD when used. The guideline has a uniform standard for recreational noise exposure, unlike occupational noise exposure, which considers each nation’s laws and industrial situations. When scrutinizing these guidelines narrowly, they seem to merely inhibit the use of PLDs and may be less persuasive for changing the voluntary usage patterns of PLD users. Thus, young people have maintained poor listening behaviors [18] and have not modified their potentially harmful behaviors, even those that might cause hearing loss [19] despite being offered several educational programs designed to prevent that hearing loss. Chung et al. (2005) reported that only 8% of young participants even recognize hearing loss as a major problem [20]. However, none of the previous studies analyzed the relationships between the use of PLDs, hearing insults, and current knowledge related to hearing conservation. These relationships are likely to result in compliance with recommendations for the safe use of PLDs, particularly among young people who may have high risk of hearing loss, which broadly includes the loss of auditory awareness in certain situations that could pose a danger to both PLD users and those around them [9].

In this light, the purpose of the present study is to analyze the experience of using PLDs among Korean college students with a large sample size. A second purpose of this study is identifying their knowledge and their attitudes toward hearing conservation. Finally, a third purpose of the present study is to offer a framework for seeking and processing hearing problems and conservation [21] that address the following two research questions:RQ1. Which behavior involved in using PLDs might lead to a self-reported and experienced hearing problem?RQ2. Does knowledge of hearing loss among PLD users influence their attitude toward hearing conservation?

We believe that the results of the current study can provide new insight into the ongoing use of PLDs among college students and motivate educational institutions to establish appropriate hearing conservation guidelines.

## 2. Materials and Methodology

This study was divided into two phases. In the first phase, we collected data using a questionnaire to analyze PLD usage and opinions related to hearing conservation in college students. In the second phase, we investigated certain relationships found in the data by using hypothesis testing and statistical validation.

### 2.1. Phase 1. Analysis of the Questionnaire

#### 2.1.1. Development of the Personal Listening Device and Hearing Questionnaire-Korean (PLDHQ-K)

The survey instrument was the Personal Listening Device and Hearing Questionnaire (PLDHQ) that Danhauer et al. developed [9]. We adapted it via translation and back-translation from English to Korean with the authors’ permission. Although the original version of the PLDHQ had 83 items, the present study only included 78 items, as 5 were removed as they had less relevance or did not fit in either the current situation or the culture in Korea. In the PLDHQ, for instance, the iPod was assumed to be the main PLD, but it has since been replaced with the Smartphone’s MP3 player function. After receiving professional input from colleagues, the Korean version of the PLDHQ (PLDHQ-K) was fully developed.

Apart from the participants’ demographic statistics, the PLDHQ-K consisted of 9 categories: (1) knowledge about hearing loss and hearing conservation, (2) self-evaluation for hearing status and the experience of noise exposure, (3) PLD preference, (4) use pattern for PLDs, (5) reasons for using PLDs, (6) hearing effect of PLD use, (7) opinion on PLD use, (8) knowledge and attitude toward regulations, and (9) preference for education. The final item was intended to be a validity check; the participants were asked whether they had given generally conscientious responses to the questionnaire.

#### 2.1.2. Participants

As the target population, we recruited Koreans, primarily in their 20s and 30s, who currently attended college in South Korea. The online survey directly contacted this population by placing advertisements on social networking sites, such as Facebook and Instagram, during spring term 2018. Participants could access the PLDHQ-K, and the advertisements offered exposure to the population 56,506 times (the response rate was 1.79%). A total of 1009 participants were asked to volunteer to complete the online version of the questionnaire. The questionnaire took into consideration the regional distribution of Korean college students in seven major provinces (e.g., Seoul, Gyeonggi-do, Gangwon-do, Chungcheong-do, Jeolla-do, Gyeongsang-do, and Jeju Island) [22]; 96.8% of the respondents were 20 to 27 years in age. Seniors had the highest number of respondents (30.9%) and freshman the lowest (14.6%). Gender was almost equally distributed with 506 males (50.1%) and 503 females (49.9%). Detailed information regarding the participants is summarized in Table 1. The survey was completed by the 1009 respondents with high reliability (Q78, 99.3% for honesty).

#### 2.1.3. Data Analysis

Participants’ responses to the 78 items were tallied using Survey Monkey, which automatically calculates with high accuracy the percentage answering in each category for each item. The analyzed data were classified into similar topics and are categorized here in Appendix A, Appendix B, Appendix C, Appendix D, Appendix E, Appendix F, Appendix G, Appendix H and Appendix I for number of respondents and percentages. Additionally, “inserted Excel worksheet object to” was added as a graphical display in each item of the appendixes for better visualization.

### 2.2. Phase 2. The Regression Model

#### 2.2.1. Research Hypotheses

To observe certain relationships closely in this large data set, we formulated two hypotheses for the current study.

**Hypothesis 1** **(H-1):**
*Of the various behaviors of PLD usage, two chronic factors, e.g., volume level and usage frequency, may negatively affect users’ self-reported and experienced hearing problems.*


**Hypothesis 2** **(H-2):**
*Our respondents’ knowledge of hearing loss helps determine their attitudes toward hearing conservation in either a positive or a negative way.*


#### 2.2.2. Statistical Analysis

To test our research hypotheses, 78 items of the PLDHQ-K were analyzed using principal component analysis (PCA) with varimax rotation as the first step. Thirteen items related to 3 factors (i.e., using time/frequency of PLD use, volume of PLD, and current hearing problems) were extracted in H-1, while another 13 items related to 4 factors (knowledge of hearing loss and 3 kinds of attitude) were tied in H-2. The selection criteria for each factor were the Kaiser’s rules, which select the factors that have eigenvalues greater than 1.0. Then, several cause-effect relationships of the items for each hypothesis were significantly explained using multiple regression analysis. All analyses were conducted for a statistical level of *p* < 0.05.

## 3. Results and Discussion

### 3.1. Phase 1. Analysis of Questionnaire

An online survey was employed to analyze the experience of using a PLD for 1009 Korean college student respondents to determine the level of their knowledge and their attitude toward hearing conservation.

#### 3.1.1. Self-Evaluation for Hearing Conditions and the Experience of Noise Exposure

In terms of self-evaluated hearing condition, 76.1% of the respondents evaluated themselves as having good hearing (Q17) and 53.5% reported having no difficulty in hearing (Q35). Also, our respondents seemed generous about rating their own hearing condition. This result is similar to the result of the PLDHQ, with only 12% of respondents having experienced hearing difficulty [9]. Nevertheless, if the participants did have a hearing problem, they attributed it to PLD use (30.4% of Q36 and 79.2% of Q65); the PLDHQ returned similar results. It is interesting to recall that Portnuff et al. (2016) showed that hearing loss can be caused by both excessive and erroneous use of PLDs [23].

For experience of noise exposure, approximately 77.2% of the respondents always/frequently/sometimes were in a noisy setting (Q5). Consistent with our expectations, many college students are susceptible to environmental noise, but ironically PLD users tend to set their device volume higher in environments where there is background noise [24]; consequently, PLD usage in the noisy environments that college students experience cannot be ignored when developing better guidelines. For additional details on this point, see Appendix A.

#### 3.1.2. Knowledge of Factors Triggering Hearing Loss and Hearing Loss Prevention

Of the 1009 respondents, 72.2% reported that hazardous noise could cause secondary hearing loss for those who already had a hearing loss (Q6). Of these respondents, 66.2% knew that the hearing loss might be preventable (Q8). This response means that many college students do understand that high levels of noise can cause damage and that it is possible to prevent noise-induced hearing loss. Similarly, 74% and 82.3% of U.S. college students responded to the same questions, respectively, in the PLDHQ [9]. Muchnik et al. (2012) also reported that 88% of the respondents had real/specific knowledge of hearing loss, and 79% knew that listening to loud music was related to hearing loss [25]. Nevertheless, our respondents did not know of any effective method of hearing conservation (Q7), so it will be necessary to provide more education about hearing conservation for better public health in the near future.

Most of the college students were aware of the symptoms related to overexposure to hazardous noise. In detail, about 80% of the respondents agreed that increasing the volume of television (TV) or a radio was a sign of hearing loss (Q15); 75% also knew that experiencing tinnitus was possible after exposure to high levels of noise (Q9). However, symptoms, such as “asking to repeat what it was” (Q11) and “speech seems to be heard as a muffled or mumbled sound” (Q13) were less familiar to these respondents. Interestingly, a regional difference from previous studies was found for how to recognize the signs of hearing loss. Two-thirds of U.S. college students [9] and half of Canadian college students reported that tinnitus was the first sign of hearing loss [6], while 60% to 70% of the students in both Mexico and Puerto Rico recognized that “turn up the volume” and “ask to repeat” were common signs of hearing loss [8]. For more detailed results, see Appendix B.

#### 3.1.3. Preference for Personal Listening Device (PLD) Use

The preference for PLD uses dealt with such as penetration rate or type of PLDs. More specifically, almost all the respondents had used PLDs (97.6%, Q18), and most of their friends also had used them (77.1%, Q22). Reflecting the current high distribution rates of smartphones in Korea [26], 93.8% used the PLD function of their smartphones (Q19). In addition, 70.2% had used a smartphone as a PLD (Q30) and 80.5% considered a smartphone to be the most preferred PLD (Q23). In sum, the high distribution rates of the smartphone may influence users to use the PLD function more than in the past, resulting in further concern about unlimited use of PLDs that can cause music-induced hearing loss [27] and simultaneously another potential risk, such as PLD oblivion [28]. With respect to obtaining a PLD, 80.3% of the college students purchased it themselves (Q20). Indeed, 65% said it was not a big deal to purchase a PLD even when considering their current financial status (Q21). Samsung was the most popular manufacturer of PLDs (51.4%) in Korea and Apple followed in second position (35.2%) (Q29), but unfortunately there was no indication of ongoing student compliance with the existing standards for the use of safe listening devices [17,29].

Here, an important, but expected, finding was that most of the college students listened to their PLDs using an earphone of the earbud type (66.2%, Q31). When combining PLDs with earbuds, the average attenuation of background noise was the lowest of other types of earphones [30] and users’ preferred listening levels were recorded approximately 20dB(A) higher in a noisy environment than in a quiet condition [24]. Remember that wearing earbuds to listen to PLDs in a noisy environment can compromise the user’s auditory system due to the increased PLD volume. Still, the respondents would consider using customized earphones or headphones to effectively reduce background noise (60.6%, Q32), but the purchase price had to be inexpensive, i.e., under $50 (80%, Q33). Clearly, many additional studies need to be conducted to determine safe volumes for PLDs for the various earphone types, particularly with background noise and daily noise exposure. For more detailed results, see Appendix C.

#### 3.1.4. Use Pattern of PLDs

Of the college student respondents, 77% had used a PLD for more than two years (Q34). Although about half of the respondents used their PLDs all days in the week (Q37), their use time per session was not long at 0.5 to 1 h (34.9%) and 1 to 2 h (29.6%, Q38). Also, the average use time of the PLDs in a day was rather widely distributed, ranging from 0.5 to 4 h (Q39). Regarding frequency of PLD use for more than 4 h in one particular place during a year, 1 to 3 times was the highest response (37.6%, Q40) and followed 21 or more times (23.4%, Q40). Interestingly, these results showed certainly divided responses in terms of using time in one session. In short, Korean college students did not seem to use PLDs for enough extended time to damage hearing, but they did listen longer than participants in the Kaplan-Neeman et al. (2017) study, which reported less than 30% using a PLD for 0.5 to 1 h in a single day [31]. However, the PLDs users such as ‘21 or more’ (23.4%, Q40) should be considered heavy PLD users and managed in terms of public health.

The response for Q41, which asked about preferred volume was widely distributed at volume steps 4 to 8, although volume step 6 was the most frequent response (19.1%). Similarly, 52% of the respondents thought their volume was only ‘medium’ (Q42). This was not a concern when compared to the results of Kaplan-Neeman et al. (2017) [30]; their respondents reported their volumes as ‘high’ (43.59%). Interestingly, Levey and Fligor (2011) found that African American PLD users listened at higher levels than white/Hispanic users do, and thus researchers should consider any ethnic differences reflected in their collected data [32].

Although a relatively small number of respondents reported “using PLDs for more than 4 h” (17.3%) (Q39) and “using PLDs 21 or more times per year” (23.4%) (Q40), these results should be observed with caution. While updating the 60–60 rule (i.e., volume of up to 60% and less than 60 minutes’ use each day), Portnuff and Fligor (2006) estimated that typical listeners could use their PLDs safely at 70% of full volume for 4.6 h a day if using the supplied earphones without greatly increasing their risk of hearing loss [33]. Nevertheless, a few college students in our study may still have been at risk when listening in noisy environments and/or for long periods of time, which indicates there is a need for more outreach directly to them.

With respect to background noise, 86.3% of the respondents reported increasing the volume steps of their PLDs (Q44), which was also supported by several previous studies in which the preferred listening levels of PLDs increased with higher background noise [10,34]. For instance, when listening to favorite music (62.3%, Q43) and exercising (58.3%, Q45), users tend to increase the volume steps of their PLDs. This finding is supported by PLDHQ research in that more than half of the students reported that they turned the volume up on their PLDs when they listened to their favorite songs and when exercising [9]. In Q46, activity involving their use of PLDs included riding public transportation (18.5%), relaxing (18.1%), and walking/jogging (17.3%). PLD users also reported feeling calmer during their commute and experiencing more pleasure during even mundane work [35]. Noise levels on subway platforms are on average 85.7 dBA, with levels in subway cars ranging from 84 to 112 dBA [36]; thus, PLD users using public transportation and walking/jogging are exposed to high levels of background noise. However, Levey et al. (2011) showed that commuters using the subway (93.3 dBA) did not have significantly higher PLD sound exposure than non-subway commuters (92.3 dBA) [31] and we expected that Korean college students will also use a relatively low listening volume because they adjust the volume of PLDs to avoid disturbing others in public (72.8%, Q74). Furthermore, the reported situations, i.e., walking, jogging, biking, and driving, may preclude an awareness of hazards like automobiles and muggers and might possibly interfere with users’ safety and that of others. These results indicate that education campaigns should include information beyond just the risks of hearing loss from PLDs [9]. For further details, see Appendix D.

#### 3.1.5. Reasons for PLD Use

For college students, the main reason for using PLDs was to listen to music (95.3%, Q68), and they considered PLD convenience to be an advantage (80.1%, Q76). Also, they used their devices when feeling bored (77.4%, Q75), to help them relax (59.9%, Q69), to help them concentrate (54.7%, Q72), and to isolate themselves from others (46.2%, Q70). Indeed, music is known to stimulate the brain and change a listener’s emotional state, causing either relaxation or excitation [37]. PLDs are also particularly appealing to people from modern affluent cultures, where there is a high premium on personal space, leading to a desire to withdraw and escape the streets. Thus, these users can escape from the uncontrollable sounds of the city, thereby avoiding car alarms, subway noise, car horns, and being asked for directions [38]. Unlike the iPod, which was the symbol of a new generation in the 1990s and a marker of social status [39], contemporary college students did not regard their PLD as either a fashion item (10.3%, Q71) or a form of rebellion (23.1%, Q73). Moreover, the PLD was no longer accepted as being high technology (52.4%, Q77). Thus our thinking went beyond music to the billion-plus videos enjoyed around the world; these videos usually employ streaming media (e.g., YouTube, Netflix), which offer a platform for people around the world to watch and share video content without any time and space constraints [40]. Therefore, future study of PLD abuse should not only focus on listening to music, but also consider various other sources and content that produce loud sounds. For detailed results, see Appendix E.

#### 3.1.6. Hearing Effects from PLD Use

Most (58.3%) of the college students did not have tinnitus (Q10). The students did not experience tinnitus (75.4%, Q49) or any other ear problem (76.2%, Q50) because of their PLD use. Nevertheless, the experience of saying “huh?” or “what?” and asking for repeats was reported by 68% of respondents (Q12), but not because of the PLDs (61.7%, Q53). Our participants did not concede any possible association between PLD use and this hearing effect. Also, our PLD users did not use them at sufficiently high intensity to undergo instant signs of hearing loss right after using the PLD. These patterns were similar to those found by Danhauer et al. [9]. In Q47, 66.5% of respondents reported that when they used PLDs, people around them could not hear their PLD sound. Also, 78.6% of the respondents said that they did not receive warnings from nearby people to reduce the volume of their PLDs (Q48). These results are similar to the responses to Q74 in that Korean college students did pay attention and avoided disturbing others. For further results, see Appendix F.

#### 3.1.7. Opinion Regarding Use of PLDs

Among the college students, 31.7% considered the acceptable age for using PLDs to be “5 to 10 years old” either 30.7% reported “11 to 13 years old” (Q54). These results are less conservative than Danhauer’s (2009) study which reported “11 to 13 years old” (39.2%) and “14 to 15 years old” (30.7%) [9]. Keep in mind also that PLD use at an early age presumably does damage the users’ hearing.

Fortunately, the students recognized that listening to PLDs at a high-intensity level can potentially cause hearing loss (90.7%, Q56). In addition to the hearing loss, students agreed that using PLDs could be dangerous in other situations, such as driving and snowboarding (83.1%, Q55), and recognized their concentration on PLDs (70.7%, Q66). Thus, we support more consciousness-raising messages that suggest that college students do not use their PLDs in situations that require their focused attention [9]. Our respondents also thought that PLDs should come with a statement that listening to music at high intensity for a long period of time may cause hearing loss (Q59). In addition, the respondents consider decreasing the listening level rather than listening time to minimize the risk to their hearing (Q60). Nevertheless, 41.5% of respondents said that they did not want to change their use patterns even if scientific evidence proved that using PLDs at high volumes can cause hearing loss (Q58). Although some warnings about music-induced hearing loss have been issued by the WHO [17], only 33.3% of our respondents considered it important to follow the manufacturer’s guidelines (Q57), a much lower number than the 61.2% of U.S. college students [9]. These results are likely due to having no short-term changes after exposure to high-intensity of PLDs [41] or having insufficient education opportunities. For detailed results, see Appendix G.

#### 3.1.8. Knowledge of and Attitude Toward Regulations and Preference Type for Education

About half of the respondents (56%) asserted that PLD manufacturers should provide a function of volume limitation for their customers. Although 65.3% of the Korean college students were not familiar with recommendations and conservation methods offered by the manufacturers (Q25), they positively responded to using volume control software given from the manufacturer (55.8%, Q26). Additionally, they agreed that the ‘60–60 rule’ (62%, Q27) was not a violation of the public’s privacy (59.4%, Q28). Although these results were similar to those of Danhauer et al. [9], the U.S. college students agreed relatively less often (24.5%) about the ‘60–60 rule’, which seems to reflect a clear cultural difference between the two countries.

Of the respondents, 60.3% thought that current media, such as news outlets and journals, did not exaggerate the possible risk of a hearing loss caused by PLD use (Q61); however, 77.4% wanted to learn about potential hearing loss caused by excessive use of PLDs (Q62). The respondents preferred to obtain that information via e-mail (23.9%), TV (22.5%), and sometimes experts (15.4%, Q63). Most of the students also wanted to be advised by a hearing professional (42%) and a doctor (36.6%) on how to prevent noise-induced hearing loss (NIHL) (Q64). Marron et al. (2015) supported this result; 84% of their respondents stated they would shorten the duration of their PLD use or reduce their listening level if given information by audiologists or doctors [42]. Thus, they indicate a willingness to change their behavior if they deemed the behavior to be unsafe. Ultimately, audiologists, physicians, and manufacturers should work together to develop effective educational outreach campaigns targeting this particular population [9]. Broader education on the appropriate use of PLDs and the effect of noise on hearing is essential [43], especially given the worldwide sales of 25.6 million portable music players in 2005, which was an increase of 409% from the previous year [44]. For detailed results, see Appendix H and Appendix I.

#### 3.1.9. Comparison of Responses for PLDHQ and PLDHQ-K

Although most of our questions received responses similar to those for the PLDHQ, four items (Q9 along with 11, 13, and 15, Q37, Q62, and Q65) showed a prominent difference between the two questionnaires, as shown here in Figure 1.

For the number of days in a week that PLDs were used, approximately 50% of the Korean students reported always using them in a week (Q37, Figure 1A). This use is relatively higher at 25.3% than for U.S. students. Also, approximately 50% of the Korean college students responded that if they had a hearing loss, its cause would be PLDs (Q65, Figure 1B), but only 38.78% of U.S. college students reported this response. This discrepancy is presumed to stem from the time difference of about 10 years between the study and the increased penetration and accessibility of PLDs. In modern life where the smartphone lets people listen to music and watch video clips anywhere. Figure 1C shows the response rate for the question about a sign of hearing loss (Q9, 11, 13, and 15). In the case of the Korean college students, increasing the volume steps of TV or radio (56.49%) and having tinnitus (47.97%) had a relatively higher response, whereas only tinnitus showed a dominant pattern over other responses (75.05%) in the PLDHQ. For the question about learning and its association with PLD use and hearing loss, Figure 1D presents a different pattern between the two groups of college students. There were more ‘yes’ responses for the Koreans and more ‘no’ responses for the U.S. college students. Zogby (2006) explained this difference between responses by suggesting that African Americans (72%) were more likely than Hispanics (56%) and whites (48%) to report that they had some symptoms of hearing loss and to know the positive methods for hearing conservation [8]. Thus, it is better to apply a different approach for effective education for each culture and country.

### 3.2. The Regression Model

The results of the Kaiser–Meyer–Olkin (KMO) and Bartlett’s tests were acceptably high at 0.835 (*p* < 0.001) for the first hypothesis (H-1), meaning the data were suited for PCA using the varimax rotation method. The eigen values were 27.796, 19.111, and 15.704 for the three factors that contributed to their relationship. Because the PCA’s results demonstrated the factors and eigen vectors for each item, linear regression analysis was applied after conducting the PCA. Table 2 shows the results of multiple regression analysis for the first hypothesis.

Figure 2 presents the results of the regression model and demonstrates that frequent use of the PLD and the high volume levels of the PLD affected the hearing condition in 1009 respondents. Although both factors, namely, usage frequency (Q 39 and 40) and level of volume (Q 42 and 48), significantly affected current hearing problems, the volume level of PLD usage, especially Q48, strongly supported hearing problems for PLD users with higher standardized regression coefficients (0.338~0.394, *p* < 0.001). Regardless, the interpretation should be different when considering the relationship between the factors and hearing problems. In terms of use time (Q39), the significance values mean the tendency of respondents that use PLDs for 0.5 to 4 h per day was less experience of tinnitus and less need to ask someone to repeat. The frequency of use was also significant and correlated with more questions of hearing problems (Q40). However, because the responses of Q40 were dichotomous regarding low-or high-use frequency, it should be carefully interpreted; the respondents used PLDs more than 4 h a day 1~3 times (36.7%) or more than 21 times (23.4%) in a year. In addition, the use time and frequency factor provided a weaker explanation at 0.072~0.127 than did the volume of PLDs. This relatively weak explanation is presumed to be due to the dichotomous response of Q40. Despite the dichotomy, the frequency of long-term use and hearing problems were not related. In other words, the use time and frequency of using PLDs among Korean college students was considered too insignificant to cause them to experience hearing problems (signs of actual hearing loss).

Although the volume of PLDs was statistically strong, it was also interpreted similarly to use time and frequency. For Q42, more than half of the respondents said that they usually set a ‘medium’ level volume. These results, understandably, are associated with less experience of hearing problems. Also, those respondents whose family, friends, or others had not warned them to turn down the volume of their PLD (Q48) were associated with less experience of hearing problems. In other words, Korean college students do not use volumes high enough to experience signs of hearing loss. As shown in Figure 2, both factors (use time and frequency and volume of PLDs) were significantly related to hearing problems. However, with more insightful interpretation of the relevance of each item, one can say that Korean college students indicated that they did not overuse PLDs until they began to experience actual hearing problems that mean signs of hearing loss such as tinnitus, increased volume, and muffled sound, especially after using PLDs.

For the second hypothesis (H-2), the KMO and Bartlett’s measure was 0.702 (*p* < 0.001). Four factors were confirmed, and their eigenvalues were 19.251, 11.607, 10.097, and 9.068. The results of multiple regression analysis for H-2 are shown in Table 3. Figure 3 demonstrates that the amount of respondents’ knowledge (Q9, 13 and 15) regarding hearing loss supports the positive and/or temperate attitudes toward hearing conservation. In detail, these positive attitudes revealed two subgroups of informational and technical restrictions. The informational restriction can be explained as positive acceptance of the capability of being restricted based on knowledge or information (Q56, 59 and 60), whereas the technical restriction focused on acceptance of the capability being restricted based on volume control function/software (Q24 and Q26). Statistically, in the Figure 3 model, the respondents who knew that they were experiencing tinnitus (Q9) or that increasing the volume steps (Q15) was a symptom of hearing loss thus preferred the technical restriction (Q24; 0.154, *p* < 0.001) and the information restriction (Q56; 0.186, *p* < 0.001), respectively, and had a positive attitude toward hearing conservation. However, the respondents of Q15 also partially supported Q61 (0.116, *p* < 0.001) with a temperate attitude toward hearing conservation.

In terms of the relationship between the amount of knowledge about hearing loss and attitudes toward hearing conservation, the more subjects there were with knowledge of hearing loss, the more positive became the attitude toward informational and technical restriction [21]. Thus, the subjects who were knowledgeable about hearing loss were receptive to restrictions. In addition, the subjects with a temperate attitude also thought that media, such as newscasts and newspapers, did not overstate the risk of hearing loss caused by PLD use. In short, Korean college students were knowledgeable about hearing loss and had a receptive attitude toward restrictions for hearing conservation.

## 4. Conclusions

The Korean college students were aware of their listening levels and partially knew the maximum amount of time that they could listen at their chosen levels without risking hearing loss. Based on these current results, educational programs should provide more information on the effective actions necessary to minimize risk for hearing loss, while also increasing the public’s knowledge and consequently changing both people’s attitudes and listening habits. It is also crucial to develop appropriate standards and safe recommendations for daily music exposure in future studies that can confirm the longitudinal effect. We believe that these studies can provide further insights into the information educational institutions need to help establish a preventive program that can target the inappropriate use of PLDs effectively.

## Figures and Tables

**Figure 1 ijerph-17-02934-f001:**
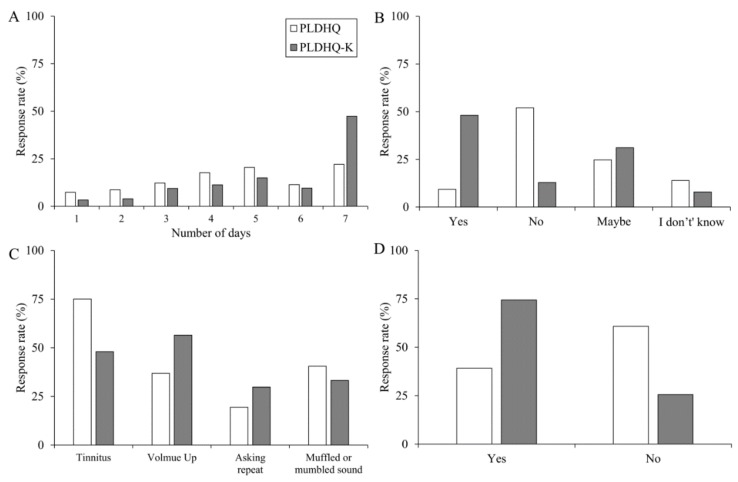
Data comparison of the original Personal Listening Device and Hearing Questionnaire (PLDHQ) and the current Korean version (PLDHQ-K): (**A**) How many days do you use your PLDs in a week? (Q37), (**B**) If you already have a hearing loss, do you think that your use of PLDs is one of its causes? (Q65), (**C**) Questions regarding signs of hearing loss (Q9, 11, 13, and 15), (**D**) Do you want to study more about the potential hearing loss associated with using PLDs? (Q62).

**Figure 2 ijerph-17-02934-f002:**
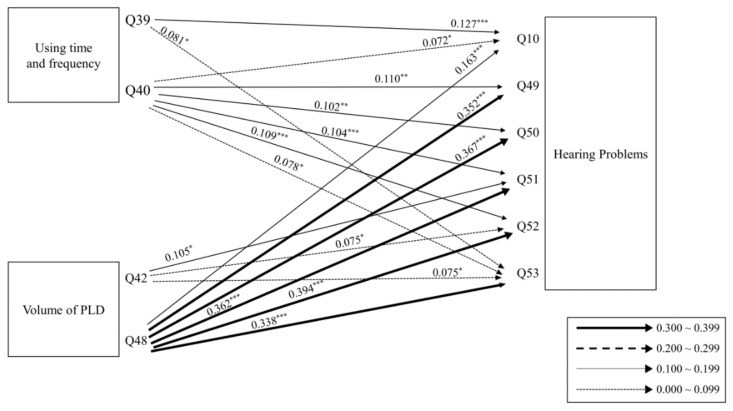
Path result for the first hypothesis based on a multiple regression model. * *p* < 0.05, ** *p* < 0.01, *** *p* < 0.001.

**Figure 3 ijerph-17-02934-f003:**
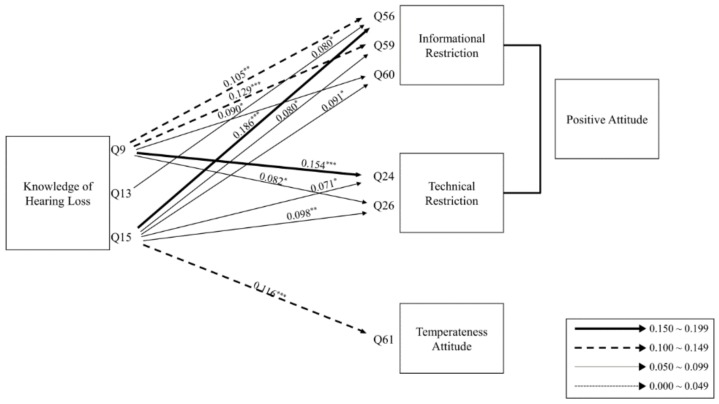
Path results for the second hypothesis based on the multiple regression model. * *p* < 0.05, ** *p* < 0.01, *** *p* < 0.001.

**Table 1 ijerph-17-02934-t001:** Demographic statistical information of survey participants.

Item Number	Response	Number of Respondents (%)	
Q1. Age	19 years and younger	18(1.8)	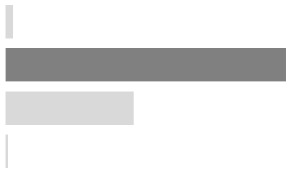
	20~24 years	674(66.8)	
	24~29 years	309(30.6)	
	30 years and older	8(0.8)	
	Total:	1009	
Q2. Gender	Male	506(50.1)	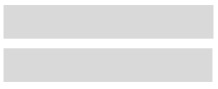
	Female	503(49.9)	
	Total	1009	
Q3. Residential District	Seoul	171(17)	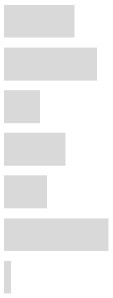
	Gyeonggi-do	25(22.3)	
	Gangwon-do	87(8.6)	
	Chungcheong-do	150(14.9)	
	Jeolla-do	105(10.5)	
	Gyeongsang-do	254(25.2)	
	Jeju Island	16(1.6)	
	N/A	1	
	Total	1009	
Q4. College Status	Freshmen	147(14.6)	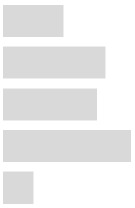
	Sophomores	250(24.8)	
	Juniors	228(22.6)	
	Seniors	311(30.9)	
	Graduate Students	73(7.2)	
	Total:	1009	
Q78. Self-assessment Validity	Completely honest	961(95.24)	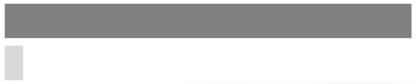
	Partially honest	41(4.06)	
	Dishonest	7(0.69)	
	Total:	1009	

Note. Items receiving a 50% and higher response rate are graphically marked in darker gray, Q: Question.

**Table 2 ijerph-17-02934-t002:** Summary of the multiple regression analysis for the first hypothesis.

Model	R^2^	Adjusted R^2^	Durbin-Watson
Q10	0.064	0.061	2.120
Q49	0.166	0.164	2.063
Q50	0.153	0.152	2.033
Q51	0.239	0.236	2.027
Q52	0.209	0.207	1.967
Q53	0.175	0.172	2.035

Note. Q: Question.

**Table 3 ijerph-17-02934-t003:** Summary of the multiple regression analysis for the second hypothesis.

Model	R^2^	Adjusted R^2^	Durbin-Watson
Q56	0.068	0.065	1.958
Q59	0.026	0.024	2.024
Q60	0.019	0.017	1.953
Q24	0.032	0.030	2.021
Q26	0.019	0.017	2.072
Q61	0.014	0.013	1.986

Note. Q: Question.

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
