# Peer review of "Use of Personal Listening Devices and Knowledge/Attitude for Greater Hearing Conservation in College Students: Data Analysis and Regression Model Based on 1009 Respondents"

_ijerph, 2020, doi:10.3390/ijerph17082934_

Round 1
Reviewer 1 Report
There has been a huge increase in the use of personal listening devices throughout the world, but particularly in developed nations. This was a nicely designed questionnaire study that helped to discover some important gaps in the knowledge of the effects of personal listening devices on hearing among college students in Korea who are users of these devices. The results appear to be statistically validated and the interpretation of the data appears appropriate. This manuscript provides an important contribution to educational needs among users of these devices.
There are occasional misspellings and the bar graphs in the appendix do not appear to line up well with the numerical data.
Author Response
Thank you very much for your valuable and significant comments. Based on the comments received, we had discussed several times and understood the reviewers’ multiple concerns. Our paper was changed into a better version while considering all comments which the reviewers pointed out. Please see our response in the table below and find newly changed parts of red letters in our revised manuscript. Misspellings and the bar graphs in the appendix were modified. Again thanks
Please see the attachment

Reviewer 2 Report
The manuscript studies Korean college students and aims at providing a qualitative analysis on their knowledge of hearing and hearing loss. The study addresses 2 questions: Which behaviour involved in using a personal listening device might lead to hearing problems? ii) Does knowledge of hearing loss influence their attitude toward hearing.
Specific comments are added in the attached pdf file.
Despite asking an interesting question, (e.g whether awareness leads to prevention of hearing loss (aim 2)), the study suffers from
- weaknesses linking to lengthy paragraphs, lacking focus, where questions are unrelated to the specific topic or questions (comments in the text).
- an absence of quantification of hearing loss or hearing deficit (objective assessment) makes it difficult to evaluate the link between knowledge of hearing loss and attitude toward hearing.
- Personal opinions or statements that weakens the assessment
- An absence of a line of research (clear hypothesis) that could have been investigated by focused questions.
- A weak conclusion with no link to the original hypothesis

Author Response
Thank you very much for your valuable and significant comments. Based on the comments received, we had discussed several times and understood the reviewers’ multiple concerns. Our paper was changed into a better version while considering all comments which the reviewers pointed out. Please see our response in the table below and find newly changed parts of red letters in our revised manuscript. The line numbering based on attached pdf files. Again thanks.
Please see the attachment.

Reviewer 3 Report
Korean college students (N = 1009) completed a 78-item survey that assessed 9 categories. Results indicated that hearing problems were related more to volume than frequency of PLD use. Given the positive attitude of students toward receiving more information, more information about hearing conservation could be helpful in restricting PLD use. The authors suggest that audiologists, physicians, and PLD manufacturers should work together to develop effective educational outreach campaigns targeting college-aged PLD users.
I have no major concerns. Most of my minor suggestions relate to grammar.
Please address this disparity: How does Q37 (half used PLDs all days in the week, L216) jibe with Q40 (23% use PLDs 21 or more times per year (L231 and Figure 1).
L26 add “than to” their frequent use.
L42. terminology (“crazy about PLDs”) is imprecise, unclear and potentially offensive
L79 add s (includes) if this clause refers to hearing loss, not recommendations
L110. “Got the final item”? Perhaps: “The final item was intended to be a validity check. Participants were asked…”
Table 1. Q78 and L126, doesn’t this item assess validity, not reliability?
L115 But: 26 participants were not in their 20s or 30s. Perhaps: “Koreans, primarily in their 20s and 30s,”. Also, change L115 to “who currently attended college”
L223 “asked”
L260, remove apostrophe (PLD’)
Figure 1. put space between “the” and “original”
L346. This is a study, but not an experiment (which would require random assignment of participants to groups).
L364, put space between “for” and “each”
Author Response

(The authors gave the same response as above.)

Round 2
Reviewer 2 Report
The authors answered the questions, and the manuscript can be accepted in its present form